# Graph Transformers for Large Graphs[*]

Anonymous[1,2,*,†], Anonymous[3,†] and Anonymous[4,†]

[1]*Anonymous*

## Abstract
Transformers have recently emerged as powerful neural networks for graph learning, showcasing state-of-the-art performance on several graph property prediction tasks. However, these results have been limited to small-scale graphs, such as ligand molecules with fewer than a hundred atoms, where the computational feasibility of the global attention mechanism is possible. The next goal is to scale up these architectures to handle very large graphs on the scale of millions or even billions of nodes. With large-scale graphs, global attention learning is proven impractical due to its quadratic complexity w.r.t. the number of nodes. On the other hand, neighborhood sampling techniques become essential to manage large graph sizes, yet finding the optimal trade-off between speed and accuracy with sampling techniques remains challenging. This work advances representation learning on single large-scale graphs with a focus on identifying model characteristics and critical design constraints for developing scalable graph transformer (GT) architectures. We argue such GT requires layers that can adeptly learn both local and global graph representations while swiftly sampling the graph topology. As such, a key innovation of this work lies in the creation of a fast neighborhood sampling technique coupled with a local attention mechanism that encompasses a 4-hop receptive field for each node, but achieved through just 2-hop operations. This local node embedding is then integrated with a global node embedding, acquired via another self-attention layer with an approximate global codebook, before finally sent through a downstream layer for node predictions. The proposed GT framework, named LargeGT, overcomes previous computational bottlenecks and is validated on three large-scale node classification benchmarks. We report a $3\times$ speedup and $16.8\%$ performance gain on `ogbn-products` and `snap-patents` compared to their nearest baselines respectively, while we also scale LargeGT on `ogbn-papers100M` with a $5.9\%$ improvement in performance.

## Keywords
graph representation learning, graph transformers, large graphs

## 1. Introduction

Transformer networks [1] have revolutionized representation learning in various domains, particularly in the field of natural language processing [2, 3, 4, 5, 6, 7, 8, 9]. Their unique ability to model intricate all-pair dependencies in sequential data (or sets of data tokens) has sparked interest in extending Transformer architectures beyond just sequential data, leading to promising research in the area of graph representation learning [10, 11, 12, 13, 14, 15, 16]. However, these advances are not without their challenges. As graph-based learning tasks grow more complex and the scales of the graph data increase, the limitations of current Graph Transformer (GT) architectures become increasingly evident [17, 18, 19].

On the other hand, traditional message-passing neural networks (MPNNs), including variants like GCN [20], GAT [21], and GatedGCN [22], function effectively on small-scale graphs such as molecular structures, operating in the order of $O(E)$, or $O(N)$ for sparse graphs [23]. However, their efficiency rapidly decreases when applied to larger graphs with even less than a million nodes. This is primarily because MPNNs consider all neighbors during their aggregation and update steps. To mitigate this, some approaches use neighborhood sampling (NS) to limit the number of sampled neighbors for each

*Woodstock'22: Symposium on the irreproducible science, June 07–11, 2022, Woodstock, NY*

[*]You can use this document as the template for preparing your publication. We recommend using the latest version of the ceurart style.

[*]Corresponding author.

[†]These authors contributed equally.

✉ kulyabov-ds@rudn.ru ( Anonymous); i.tiddi@vu.nl ( Anonymous); Manfred.Jeusfeld@acm.org ( Anonymous)

🌐 https://yamadharma.github.io/ ( Anonymous); https://kmitd.github.io/ilaria/ ( Anonymous);
http://conceptbase.sourceforge.net/mjf/ ( Anonymous)

🆔 0000-0002-0877-7063 ( Anonymous); 0000-0001-7116-9338 ( Anonymous); 0000-0002-9421-8566 ( Anonymous)

node up to a certain number of hops [24]. While this method keeps computational costs in check, it encounters intractability issues when the graph scales to hundreds of millions of nodes or more [25, 26, 27]. Moreover, even if NS is computationally feasible for two or three hops, the MPNNs become confined to capturing only highly localized information, which might be insufficient for tasks on large graphs where more global context is crucial [28, 29, 19].

Graph Transformers (GTs) could provide a potential solution, given their ability to model long-range dependencies and attend to global neighborhoods [15]. However, the intractability remains, given that GTs with all-pair attention, *i.e.*, each node attending to every node, would be quadratic ($O(N^2)$) computationally which is entirely feasible for large graphs. When applied with NS, GTs inherit the same limitations, confining their effectiveness to localized regions. Without NS, the task of attending to global neighborhoods necessitates approximation [30, 31, 16] or sampling [17, 32, 33] for computational feasibility, thus bringing us back to the original challenges of sampling and (in)efficient access to global information.

**Present Work.** In this paper, we introduce a comprehensive approach to overcoming the afore-mentioned critical challenges in learning on large-scale graphs by focusing on two essential design principles: model capacity and scalability. We enhance model capacity by integrating both local and global graph information for building node representations, while ensuring scalability with an efficient sampling approach. The proposed framework is summarized as follows:

1. **Framework Design**: We present LargeGT, a new framework that integrates both local and global graph representations while minimizing the computational cost incurred at both the stages. In consistency with recent working recipes in graph learning [15, 19] LargeGT utilizes two distinct modules — LocalModule and GlobalModule — to handle local and global information exchange efficiently.

2. **Localized Representations**: Within the LocalModule, we present a novel tokenization strategy that prepares a fixed set of tokens for each graph node to be processed by a Transformer encoder, resulting in rich local feature representations. Importantly, this mechanism leverages a neighborhood sampling approach that consists of an offline sampling stage and incorporates local context features, enabling a broad 4-hop receptive field through just 2-hop operations.

3. **Global Representations**: For the GlobalModule, we implement an approximate codebook-based approach, adapted from Kong et al. [19], to enable global graph attention with computational complexity linear to the codebook size. This design choice ensures that both modules can operate independently of the graph size, thereby ensuring scalability.

4. **Computational Efficiency**: Our approach effectively mitigates computational bottlenecks tradi-tionally associated with sampling techniques and global information flow. As a result, we enable incorporation of both local and global graph representations without compromising performance.

5. **Empirical Validation**: We validate the competitiveness and scalability of LargeGT with baselines in a scalable setting using `ogbn-products`, `snap-patents` and `ogbn-papers100M` datasets which are among the largest benchmarks with node in ranges 2.5M, 2.9M and 111.1M, respectively. Notably, we obtain a $3\times$ speedup and $16.8\%$ performance gain on `ogbn-products` and `snap-patents` compared to their best baselines respectively, while on `ogbn-papers100M` we scale LargeGT with a $5.9\%$ improvement in performance.

Overall, our work not only investigates on the key challenges and presents design elements for GTs at scale but also provides a robust framework for future research in this area.

## 2. Related Work

**Challenges in MPNN Scaling.** When learning on large graphs, the principal issue faced by message-passing based graph neural networks (MPNNs) is the neighbor explosion phenomenon [24, 17], since

the neighborhood sets of nodes at successive hops expand exponentially [34]. Early efforts in scaling MPNNs employed the use of neighbor sampling (NS) to sample neighbors of nodes in a graph recursively that reduces the overall neighborhood sets sending messages for nodes' feature updates [24]. While this brings a reduction in memory and compute footprint, it remains intractable to (i) aggregate information at hops greater than 2 or 3 in very large graphs due to the fact that the the size of the successive neighborhood grows exponentially, and (ii) access global information in the graph, which is also a well-acknowledged limitation for several works along this line [35, 36, 37, 38]. Information propagation *prior to* or *after* the training stage [39, 40, 41] are also adopted as ways to address the intractability brought by neighborhood explosion, which follow the aforementioned limitations. Several works also propose pre-training [27], condensation [42] or distillation strategies [25, 26] to mitigate these impacts.

**Graph Transformers and Scalability.** The apparent access to global information is a driving factor behind the recent plethora of works on Graph Transformers (GTs) with all pair attention [14]. We refer to Müller et al. [43] for a detailed taxonomy and component-wise study of GTs. However, an obvious barrier for GTs to scale to large graphs is the quadratic complexity brought by full-graph attention, *i.e.*, $O(N^2)$, with $N$ being the number of nodes in a graph. The use of sparse Transformers [15, 16] or approximated global attention [30, 31] alleviate this issue to some extent to bring down the complexity to sub-quadratic. Yet, these methods remain unscalable on single large graphs due to the entire graph structure being operated upon in memory. In order to address this limitation, either an NS-like computational boundary is enforced for each node [44, 17], or a fixed length sequence or set is prepared for each node prior to training [10, 18]. Such solutions either inherit the limitations of NS as discussed above, or are infeasible due to adjacency matrix multiplications as the graph size grows larger.

**Clustering based GTs.** Orthogonally, several recent works use hierarchical clustering or partitioning to perform global attention on the coarsened or super nodes [32, 33]. However, the coarsening step remains intractable for very large graphs with sizes in hundreds of millions or more. Finally, Kong et al. [19] use a combination of NS based local module and a global module consisting of a trainable fixed-sized codebook that represents global centroids. While the sampling limitations remain, the global module based on the centroids is efficient and something that we consider in our proposed approach to compute global representations.

## 3. Recipe for Building Transformers for Large Graphs

In the previous sections, we identified critical limitations in existing graph learning models, specifically their inability to effectively merge local and global graph features when working with very large graphs. While MPNNs struggle with the 'neighbor explosion' problem, making it difficult to aggregate information beyond 2-3 hops, GTs *can* capture global context, but are hampered by quadratic complexity in full graph attention. These challenges, although formidable, outline the essential criteria for a successful GT model tailored for large graphs. This leads us to present a recipe focusing on model capacity and scalability for building GTs for large graphs. In this section, we first discuss the design principles of model capacity[1] and scalability. These factors play a crucial role in determining the design and feasibility of a Graph Transformer, particularly when it comes to managing extremely large graphs that have node counts in the millions or higher. Finally, following the design characteristics, we introduce our proposed framework — LargeGT.

### 3.1. Design Principles

**D1- Model Capacity.** A graph learning model should possess the ability to incorporate both local and global information from the original large graph.

*Local Inductive Bias*: A node's local connectivity presents a rich source of information that is essential to utilize even in a large graph setting. A straightforward local aggregation of all neighboring nodes

---

[1]Note that in this work, we do not contextualize a model's capacity in terms of Weisfeiler-Leman (WL)'s expressiveness [45, 46, 47, 48] as it is known that almost all of non-isomorphic graphs (or subgraphs) are distinguishable by a 1-WL equivalent model (MPNN) in the presence of node features [49], and we do not consider graphs with anonymous nodes.

followed by update equation in an MPNN is impractical due to which several works utilize sampling techniques (as discussed in Section 2). The incorporation of such local information, while remaining in a feasible computational boundary, is a key ingredient when building a graph learning model for larger graphs and helps immensely in homophilic tasks [50, 51].

*Access to Non-Local Information*: A node's capability to incorporate features beyond near-local neighbors may be vital for long-range or non-homophilic tasks [28, 29]. The sampling strategies discussed in Section 2 fail to allow a model to access nodes' distant hops or global information, motivating the use of GTs [15, 16]. GTs with all-pair attention alleviate this concern, however not all such mechanisms reviewed in Section 2 are scalable, which is a key concern. The recent global attention mechanism proposed in GOAT [19] uses dimensionality reduction scheme serving as a "conceptual" global context for a node to attend when looking for non-local information. Without exhaustive computation, a large graph learning model should allow access to global graph context [52].

**D2- Scalability.** Learning on large graphs become infeasible for several existing models due to hindrances in sampling mechanisms or global information modules [14, 15, 17]. As such, we consider the following factors to be vital for computational feasibility and ensuring scalability of the GTs when graphs grow larger.

*Efficient Neighbor Node Set Retrieval*: As reviewed in Section 2, despite the use of sampling techniques, it becomes intractable to retrieve neighbor node sets from hops greater than 2 or 3 if the graph in consideration is very large [25, 26]. In fact, even a 3 hop neighbor set retrieval takes a significant amount of time for graphs with hundreds of millions of nodes, as the computational complexity of retrieving a node's $l$ hop neighborhood is $O(d^l)$, where $d$ is the average node degree. Therefore, for efficient retrieval for large graphs, we establish *only* two hops retrieval as a key constraint; this limit is commonly adopted in large-scale graph learning applications [53, 54, 55, 56].

*Efficient Global Information Access*: As much as the significance of global information is discussed as a key recipe in the Design **D1**, we reiterate that the global information flow should come at an inexpensive cost. There are multiple candidates for an efficient access to global information, such as sparse global attention [16], use of virtual nodes [52] and use of an approximate centroid-based codebook [19]. Since we focus on large graphs where the former two candidates can be infeasible due to their dependency on the number of nodes in the graph, we aim for incorporating global information without bottlenecks due to a graph's large size, such as the method implemented in Kong et al. [19] which depends on a dynamic codebook comprising of global centroid tokens.

*Distributed Training*: Handling very large graphs will be impractical without taking advantage of the distributed computing infrastructure in which several of the real world graphs occur, *eg.* social networks, which are massive and require distribution over different machines [?, 57, 58]. Ensuring efficient run times during both training and inference is vital to allow a GT to handle large graphs within acceptable time frames, keeping the cost feasible and lower. For a model to scale on large graphs, the training steps should do away with the bottlenecks brought by traditional MPNNs and sparse GTs. For instance, MPNNs usually sample nodes and their neighbors during the mini-batching step of the training process, which also applies in recent GTs [17, 19]. In principle, this translates to maintaining the adjacency matrix of a graph in one machine. This can be challenging: for example, a simple 3-layer GraphSAGE with NS requires around 350GB+ RAM on the 111.1M sized `ogbn-papers100M` following standard configurations of OGB [59]. The single-machine memory requirement can be much larger than the availability on standard machines with sophisticated models and on industry-scale graphs with billions of nodes and edges [60, 54]. Thus, avoiding the need to maintain the entire graph data on a single machine when learning is desirable.

## 3.2. Our Proposed Framework: LargeGT

Incorporating these desiderata jointly, we next introduce our proposed framework, LargeGT, which is designed from first-principles to enable the application of GTs to massive large-scale graphs. We refer to Figure 1 for a sketch of the proposed architecture.

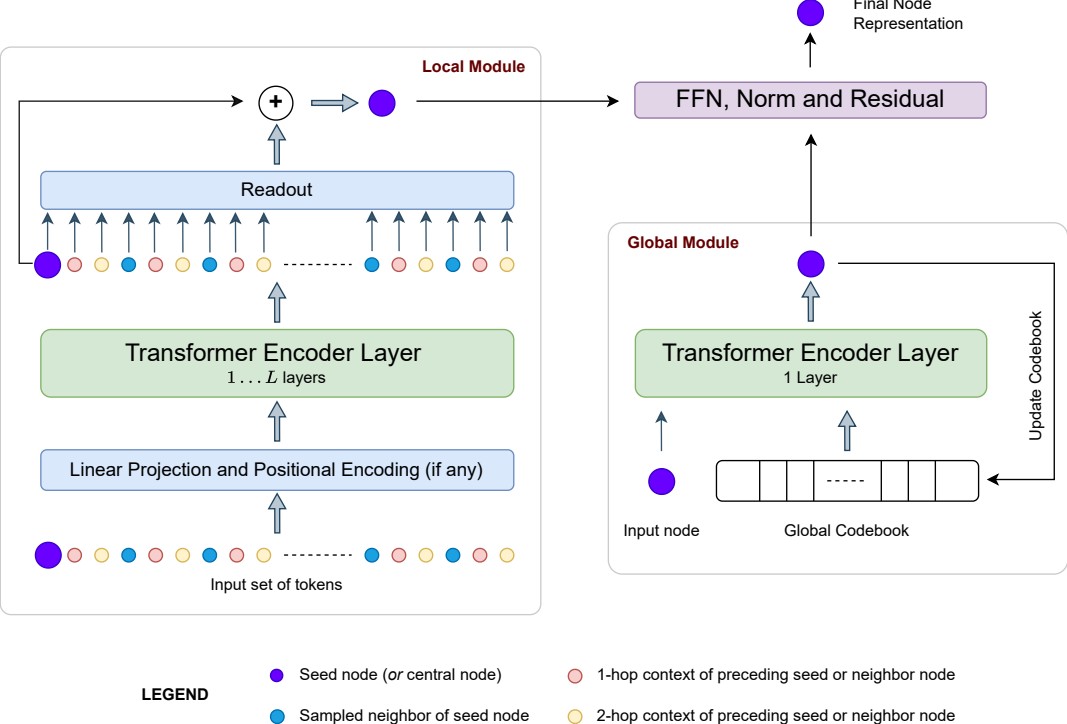

**Figure 1:** Architectural diagram of LargeGT illustrating the process of updating a node's representation denoted by the seed node. FFN and Norm denotes Feed Forward Network and Normalization layer respectively. Neighbors are sampled for a central or seed node offline, prior to the training stage using Algorithm LocalNodes. The feature vectors coming from both local module and global module are concatenated before passing to the 'FFN, Norm and Residual' module.

**Notations.** We denote a given graph with $\mathcal{G} = (\mathcal{V}, \mathcal{E})$ with $\mathcal{V}$ being the set of nodes and $\mathcal{E}$ the set of edges, and $N = |\mathcal{V}|$ and $E = |\mathcal{E}|$ being their cardinalities. The graph structure is represented by the adjacency matrix $\mathbf{A} \in \mathbb{R}^{N \times N}$ where $\mathbf{A}_{ij} = 1$ if there exists an edge between nodes $i$ and $j$, otherwise $\mathbf{A}_{ij} = 0$. The features for a node $i$ is denoted by $\mathbf{H}_i \in \mathbb{R}^{1 \times D}$ and for all nodes in the graph $\mathcal{G}$ is denoted by $\mathbf{H} \in \mathbb{R}^{N \times D}$. While we assume, for simplicity, that the graph $\mathcal{G}$ has no edge features, these can be incorporated using standard methods as done in the literature, such as using edge features during message passing to update node representations.

**Update Algorithm.** We now define the equations which update the feature representations of a node using the LargeGT framework. Given a node $i$ with its input feature $\mathbf{H}_i^{\text{in}} \in \mathbb{R}^{1 \times D_{\text{in}}}$, the aim is to obtain its output features $\mathbf{H}_i^{\text{out}} \in \mathbb{R}^{1 \times D_{\text{out}}}$ which can be passed to appropriate prediction heads and/or loss functions, subsequently, depending on the learning task. For simplicity, we will represent $\mathbf{H}_i \in \mathbb{R}^{1 \times D}$ in the following equations.

$$\mathbf{S}_i = \text{LocalNodes}_i(\mathbf{A}, K) \qquad \in \mathbb{R}^{1 \times K} \tag{1}$$

$$\mathbf{X}_i = \text{InputTokens}_i(\mathbf{S}_i, \mathbf{H}_i^{\text{in}}, \mathbf{C}_i) \quad \in \mathbb{R}^{1 \times 3K \times D} \tag{2}$$

$$\mathbf{H}_i^{\text{local}} = \text{LocalModule}(\mathbf{X}_i) \qquad \in \mathbb{R}^{1 \times D} \tag{3}$$

$$\mathbf{H}_i^{\text{global}} = \text{GlobalModule}(\mathbf{H}_i^{\text{in}}) \qquad \in \mathbb{R}^{1 \times D} \tag{4}$$

$$\hat{\mathbf{H}}_i = \text{FFN}(\mathbf{H}_i^{\text{local}} \,||\, \mathbf{H}_i^{\text{global}}) \qquad \in \mathbb{R}^{1 \times D} \tag{5}$$

$$\mathbf{H}_i^{\text{out}} = \mathbf{H}_i^{\text{in}} + \text{Norm}(\hat{\mathbf{H}}_i) \qquad \in \mathbb{R}^{1 \times D} \tag{6}$$

where, $\mathbf{S}_i$ reflects the $K$ sampled local nodes from LocalNodes in Algorithm 1, and $\mathbf{X}_i$ is the set of input tokens of size $3K$ prepared using InputTokens in Algorithm 2. Additionally, LocalModule consists of a standard Transformer encoder [1] which could be a stack of multiple layers to produce each token's representations, with a readout function at the end that converts the set of tokens to

---

**Algorithm 1** LOCALNODES: Algorithm to fetch a multiset of local nodes from 1 and 2 hop neighbors for each node.

---

**Require:** A graph with adjacency matrix $\mathbf{A} \in \mathbb{R}^{N \times N}$, and the size of the multiset $K$.
**Ensure:** Return the matrix with $K$-sized multisets for each node $\mathbf{S} \in \mathbb{R}^{N \times K}$

1: **Initialize:** Multisets for all nodes $\mathbf{S} \in \mathbb{R}^{N \times K}$
2: **for** $i = 0$ to $N - 1$ **do**
3:    $\hat{\mathbf{T}} \leftarrow 1$ and 2 hop neighbors of node $i$
4:    **if** $|\hat{\mathbf{T}}| \geq K - 1$ **then**
5:       $\mathbf{T} \leftarrow$ Randomly sample $K - 1$ nodes from $\hat{\mathbf{T}}$
6:    **else if** $|\hat{\mathbf{T}}| < K - 1$ and $|\hat{\mathbf{T}}| > 0$ **then**
7:       $\mathbf{T} \leftarrow$ Randomly sample from $\hat{\mathbf{T}}$ with replacement to make $K - 1$ nodes
8:    **else**
9:       $\mathbf{T} \leftarrow$ Randomly sample $K - 1$ nodes from $\{0, 1, 2, \ldots, N - 1\}$
10:    **end if**
11:    $\mathbf{S}_i \leftarrow \{i\} + \{\mathbf{T}\} \in \mathbb{R}^K$
12: **end for**

---

**Algorithm 2** INPUTTOKENS: Algorithm for Mini-Batch Preparation for Local Module

---

**Require:** Mini-batch of $M$ samples $\mathbf{S} \in \mathbb{R}^{M \times K}$ where $K$ is the total size of the multiset of nodes for each node, Feature matrix $\mathbf{H} \in \mathbb{R}^{N \times D}$, Hop context features $\mathbf{C} \in \mathbb{R}^{N \times 2 \times D}$.
**Ensure:** Return the input data of all nodes in mini-batch $\mathbf{X} \in \mathbb{R}^{M \times 3K \times D}$.

1: **Initialize:** $\mathbf{X} \in \mathbb{R}^{M \times 3K \times D}$
2: **for** $i = 0$ to $M - 1$ **do**
3:    **for** $j = 0$ to $3K$ with step 3 **do**
4:       $\mathbf{X}_{i,j} \leftarrow \mathbf{H}[\mathbf{S}_{i,j}]$        # node feature for node $i$ from the feature matrix $\mathbf{H}$
5:       $\mathbf{X}_{i,j+1} \leftarrow \mathbf{C}[\mathbf{S}_{i,j}, 0]$    # 1 hop context feature for the node $i$ from $\mathbf{C}$
6:       $\mathbf{X}_{i,j+2} \leftarrow \mathbf{C}[\mathbf{S}_{i,j}, 1]$    # 2 hop context feature for the node $i$ from $\mathbf{C}$
7:    **end for**
8: **end for**

---

one feature vector as sketched in Figure 1, and GLOBALMODULE consists of a single layer Transformer encoder adapted from [19] that allows a node to attend to an approximate global representation of all nodes in the graph through a projection of all nodes' features in a codebook of a fixed size, that is updated at each iteration (see Section C). $||$ denotes concatenation. $\mathbf{C} \in \mathbb{R}^{N \times 2 \times D}$ is the context feature matrix which provides the 1 and 2 hop neighborhood context for all nodes in the graph and can be precomputed as $\mathbf{C}^0 = \tilde{\mathbf{A}}\mathbf{H} \in \mathbb{R}^{N \times 1 \times D}$ and $\mathbf{C}^1 = \tilde{\mathbf{A}}^2\mathbf{H} \in \mathbb{R}^{N \times 1 \times D}$ where $\tilde{\mathbf{A}}$ is the normalized adjacency matrix [18].

Finally, FFN refers to a feed forward network used in Transformers, and NORM is normalization, which can be either of LayerNorm [61] or BatchNorm [62]. Additionally, we note here that the Algorithms 1 and 2 are defined for either all nodes or a mini-batch of nodes, while their respective outputs can be accessed index-wise in the above Eqns. 1 and 2.

**Offline Step Prior to Training.** The Algorithm 1 LOCALNODES is run on CPU prior to the training to sample local nodes for each node in the graph, and can be parallelized to multiple cores or machines. The parallelization is possible due to the fact that the fetch of 1-hop and 2-hop neighbors in Step 3 and the subsequent steps of Algorithm 1 is independent for each node and can be executed on different CPU cores in parallel. It's also worth mentioning that the graph is not required to be stored on a single machine's memory for this step. In line with our design principles, the graph can be distributed across multiple machines using techniques such as key-value stores or graph databases to circumvent the issue of memory limitations. This ensures that the offline step aligns with the scalability considerations of handling very large graphs.

**Mini-Batching for Local Module.** The Algorithm 2 abstracts a mini-batching stage during the training, which prepares the input tokens for each node in the mini-batch with $M$ node samples. This process contains a nested loop that can also be heavily parallelized across available cores. The mini-batch algorithm is implemented in a dataloader's preparation function, such as the PyTorch DataLoader collate [63][2]. Besides, an important feature provided by INPUTTOKENS is that it *increases* the receptive field of a node *up to 4 hops*, based on just 2-hop computations. To illustrate this, consider a node with its $K$ sized sampled set of 1-hop and 2-hop neighbors. Since the steps 5-6 in Algorithm 2 allows for a nodes' 1-2 hop neighbors to retrieve *their* 1-hop and 2-hop neighborhood context, the node $i$ in consideration will have access to information from up to 4 hops. We have included further elaboration of this feature in Figure B.2 in Section B.

**Characteristics of the Framework.** The LargeGT framework fulfills the desired characteristics (D1-D2, Section 3.1) for a graph learning model to scale on large graphs.

**D1-Model Capacity**: By incorporating both a local and global module, LargeGT can build node representations using information from local neighborhoods as well as global graph context. The local module leverages offline neighbor sampling and local context features to efficiently aggregate local structure up to 4 hops while the global module allows attending to the entire graph through a trainable codebook.

**D2-Scalability**: The framework only samples up to 2-hop neighbors for each node, ensuring efficient neighbor set retrieval. The runtime is also efficient as the local module operates on sampled neighbors and the global module utilizes a fixed size codebook, which is already efficient as demonstrated in Kong et al. [19]. Similarly, the offline neighbor sampling step allows the local node sets and neighborhood contexts to be prepared independently of the graph structure, essentially converting the graph learning task into a standard neural network training problem on the sampled nodes and contexts. This allows easy parallelization across machines like for other modalities such as image and text, alleviating bottlenecks caused in traditional GNN training that require adjacency matrix access on each machine, in principle. The input token preparation is also independent of the graph structure. In summary, LargeGT satisfies the key design principles of model capacity and scalability to effectively scale on large graph datasets. The local and global components allow it to learn both from local structure as well as model global dependencies in the graph, without any *explicit* bottleneck caused due to the size of a graph. In the next section, we will demonstrate that this model is not only scalable, but offers strongly competitive performance.

**Complexity.** The computational complexity of the LOCALMODULE in LargeGT is $O((3K)^2)$, while that of the GLOBALMODULE is $O(B)$ where $K$ is the size of the local nodes, $3K$ is the size of the tokens for the LOCALMODULE and $B$ is the size of the codebook in GLOBALMODULE. As such, the framework is not bottlenecked with the size of nodes in the graph, as in prior literature. Note that the complexity of Algorithm 1, which is a one-time offline step, does not affect the computational complexity of LargeGT during training or inference.

## 4. Experiments

### 4.1. Datasets and Experimental Setup

**Datasets.** We evaluate the proposed LargeGT model architecture on single large graph datasets with

---

[2]https://pytorch.org/docs/stable/data.html#working-with-collate-fn

**Table 1**
Summary of the datasets used in our experiments.

| Dataset Name | Total Nodes | Total Edges | Node Feats | Class Size | Class Label |
|---|---|---|---|---|---|
| ogbn-products | 2,449,029 | 61,859,140 | 100 | 47 | product category |
| snap-patents | 2,923,922 | 13,975,788 | 269 | 5 | time granted |
| ogbn-papers100M | 111,059,956 | 1,615,685,872 | 128 | 172 | subject area |

node classification tasks. Since we particularly focus on large graphs, we do not conduct evaluations on smaller benchmarks with lesser than million nodes as the limitations of scaling a model are not present. We use `ogbn-products` and `snap-patents` for our model prototyping and experiments, while we also scale our model on `ogbn-papers100M`. The datasets' summary is presented in Table 1.

**`ogbn-products`** is a dataset containing Amazon co-purchasing network [64] with nodes representing products and edges representing the products being purchased together. We use the Open Graph Benchmark [59] version of the dataset with their standard splits and available features.

**`snap-patents`** is a network of US patents [65] where each node represents a patent and an edge represents patent nodes which cite each other. We use the `snap-patents` dataset from Lim et al. [28] with their default splits and features.

Finally, we use **`ogbn-papers100M`** dataset [59] which is one of the largest publicly available single large graph benchmarks. Nodes in the graph denote an arXiv paper while directed edges denote papers which cite other papers [66]. As with the former dataset, we use the default splits and features provided by OGB. Among the three datasets, `snap-patents` is a non-homophilic dataset, while the rest are homophilic.

**Scalable Baselines.** We design our experiments in a way to ensure scalability with efficient neighbor node set retrieval (**D2**, Section 3.1) on large graphs, with sizes even beyond the benchmarks we use for the demonstration in this work. As such, we use constrained versions of existing MPNNs or GT baselines, denoted by Model-$\delta$ and call them 'Scalable Baselines'. We select GraphSAGE [24], GAT [21], GT-sparse [44, 11], NAGphormer [18] and GOAT [19] as the baselines which encompass fundamental GNNs as well as recent scalable GTs. Following **D2**, we constrain the graph propagation related operations in all these baselines to 2 hops only. The goal of our experimental setup is to show how the scalability constraints affect existing models' capabilities, which can be addressed by LargeGT; for this reason, we do not use enhanced techniques for obtaining top leaderboard results such as auxiliary label propagation [44, 67] or augmentations [68]. We apply similar restrictions in our proposed LargeGT as well. For hyperparameters, architectural and training setup of the baselines, we adopt the hyperparameters available in the original model papers or the OGB examples repository (see Section E). We refer to Section D for a comparison of scalable and original baselines.

**LargeGT Models.** We train and evaluate two versions of LargeGT in our experiments, denoted as LargeGT-local and LargeGT-full. In local version we omit the GLOBALMODULE (Eqn 4) and only use local representations in Eqn. 5. In the full version, we use both the local and global modules and follow the entire formulation as in Eqns. 1-6. We implement single layer of Transformer encoder in the local

**Table 2**

Results for `ogbn-products`, `snap-patents` and `ogbn-papers100M` datasets. All results reported are on 4 runs. LargeGT as well as models with -$\delta$ suffix are the versions of the respective original architecture with **D2**, *i.e.*, only upto 2-hop computations. GraphSAGE, GAT and GT-sparse use NS with sizes [20, 10]. Colors denote **First**, **Second** and **Third**. Higher is better.

| (a) ogbn-products | | | (b) snap-patents | |
|---|---|---|---|---|
| **Model** | **Test Acc** | | **Model** | **Test Acc** |
| GraphSAGE-$\delta$ | 76.62±0.93 | | GraphSAGE-$\delta$ | 48.43±0.21 |
| GAT-$\delta$ | 77.38±0.59 | | GAT-$\delta$ | 45.92±0.22 |
| GT-sparse-$\delta$ | 60.76±0.00 | | GT-sparse-$\delta$ | 47.81±0.00 |
| NAGphormer-$\delta$ | 75.28±0.04 | | NAGphormer-$\delta$ | **60.11±0.05** |
| GOAT-local-$\delta$ | **81.17±0.12** | | GOAT-local-$\delta$ | 40.95±0.16 |
| GOAT-global-$\delta$ | 70.28±1.95 | | GOAT-global-$\delta$ | 42.65±0.07 |
| GOAT-full-$\delta$ | **79.88±0.20** | | GOAT-full-$\delta$ | 50.28±0.14 |
| LargeGT-local | **78.95±0.80** | | LargeGT-local | **68.19±3.11** |
| LargeGT-full | **79.81±0.25** | | LargeGT-full | **70.21±0.12** |

| (c) ogbn-papers100M | |
|---|---|
| **Model** | **Test Acc** |
| GOAT-full-$\delta$ | 61.12±0.10 |
| LargeGT-full | **64.73±0.05** |

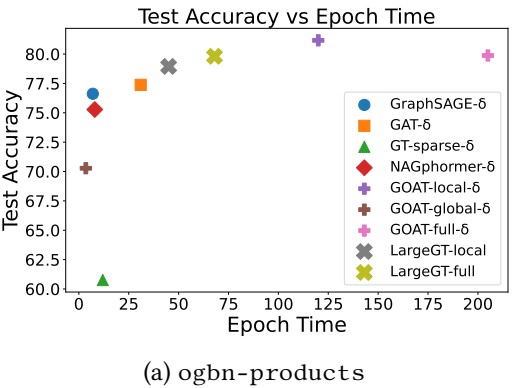
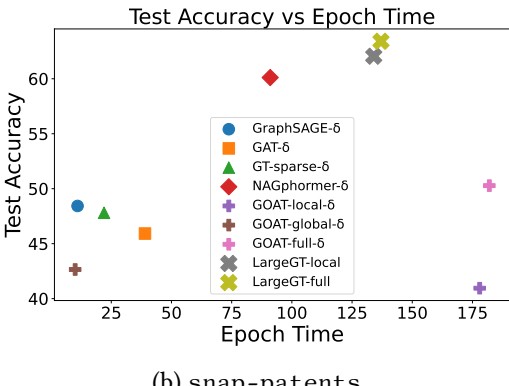

(a) `ogbn-products`                         (b) `snap-patents`

**Figure 2:** Training time per epoch of various baselines with the proposed architecture. LargeGT reports relatively faster epoch times compared to the best baselines despite performing competitively.

and global modules.[3] Other hyperparameter details are included in Section E.

**Hardware.** The experiments on `ogbn-products` and `snap-patents` are conducted on Tesla-V100 GPU with 16GB GPU memory, 24 CPU cores and 156GB RAM, while that of `ogbn-papers100M` are on A40 GPU with 48GB GPU, 128 CPU cores and 1024GB RAM.

## 4.2. Numerical Results and Discussion

We now present the numerical results in this section. Table 2 presents the main comparison of the baselines Model-$\delta$ with the LargeGT models. For `ogbn-papers100M`, we only report the best performing baseline on the homophilic task of `ogbn-products` due to computational constraints. Figure 2 shows the training time per epoch incurred by the models, while Figure 3 shows the sensitivity analysis of the number of nodes sampled ($K$) in Algorithm 1 which is used by LocalModule in LargeGT. We present our observations as follows.

**On Performance.** On comparison with the scalable baselines in Table 2 consisting of both GNNs and GTs, we observe LargeGT to attain competitive performance to the best baseline GOAT-local-$\delta$ on `ogbn-products`, while on `snap-patents`, LargeGT beats all the baselines. The lower scores of GraphSAGE-$\delta$ and GAT-$\delta$ models reveal, in part, how the constraint **D2** brings down the model capacity and such a network with a 2-hop graph receptive field may not incorporate necessary information required to build useful node representations. Among all the models on `ogbn-products`, GOAT-local-$\delta$ is the best performing, which on `snap-patents` ranks the lowest, understandably due to the latter's non-homophilic characteristic where local-only information aggregation may not be enough. All results considered, LargeGT ranks among the best models on both the datasets, suggesting the capacity to work well in both homophilic and non-homophilic settings. Finally, we report the results of LargeGT-full on `ogbn-papers100M` together with GOAT-full-$\delta$ which is the best performing baseline on a similar homophilic task `ogbn-products` on a computational budget of 48h. We observe LargeGT-full to be significantly better (5.9%) in performance compared to GOAT-full-$\delta$, hence showing the ability to scale to massive graphs.

**On Runtime.** In Figure 2, we present the training time per epoch of all the models considered. We observe the epoch time of LargeGT to be faster up to a maximum of 3× times (GOAT-full-$\delta$ vs. LargeGT-full). Further, Fig. A.1 in Appendix Sec. A shows accordingly better learning and generalization curves of our proposed architecture against the baselines.

**On Design Characteristics.** The scores of LargeGT indicate the need to incorporate both local and global neighbor information, as well as the suitability of such architecture to perform both in homophilic and non-homophilic tasks. The absolute gain in performance of LargeGT-full against LargeGT-local which is around 1% and 2% respectively for `ogbn-products` and `snap-patents` demonstrates how

---

[3]We observed decreased performance when stacking multiple layers, in consistency with recent works such as Chen et al. [18], Kong et al. [19], and leave the exploration of multi-layered version for later.

LargeGT satisfies **D1** design characteristic. **D2** which necessitates the computational feasibility of a network helps keep the epoch times of each baseline feasible, along with LargeGT. See the epoch times of each experiments in Figure 2. If we further compare GOAT-full-**δ** with the original GOAT-full (NS of 3 hops) [19], we report per epoch times of 205s and 497s respectively on `ogbn-products`, which suggests how existing works without **D2** can be computationally expensive. Finally, we ensure parallelizability in principle given that the Algorithms 1-2 effectively convert the training problem on a single large graph to a standard neural network training on the sampled tokens, which can leverage parallelization without the need of a single global graph context maintained in the memory of any one machine.

**On Node Set Sizes** $K$**.** In Figure 3, we show results of LargeGT-full experiment on an important hyperparameter in our proposed framework, the number of nodes to be sampled from LocalNodes (Algorithm 1). The value of $K$ is chosen from a list of [20, 40, 50, 60, 80, 100, 150, 200]. We observe best performance on $K = 100$ and $K = 50$ for `ogbn-products` and `snap-patents` respectively, which somewhat aligns with the extent of their tasks being non-homophilic since a lower $K$ would reflect lesser tokens from the local neighborhood.

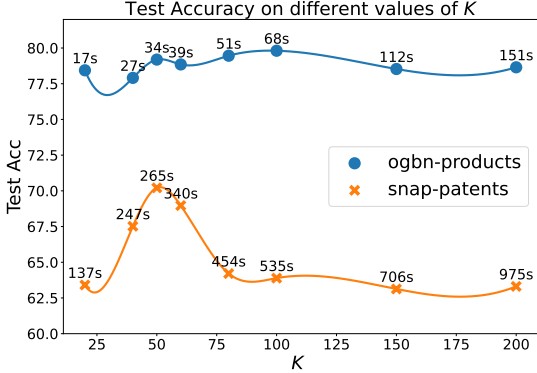

**Figure 3:** Hyperparameter analysis for $K$ which is the number of nodes to be sampled in LocalNodes algorithm for LargeGT-full model on `ogbn-products` and `snap-patents` datasets. Labels on the data points denote the training time per epoch.

**On Comparison with SGC and SIGN.** We also compare LargeGT with SGC (Simplified Graph Convolution) [40] and SIGN (Scalable Inception Graph Neural Networks) [41], which utilize simplified information propagation operations based on graph adjacency matrices. While SGC has been shown to match or marginally improve upon the performance of GCN [20] in their experiments, its scores are surpassed by those of SIGN. The SGC model is demonstrated to be a special case of SIGN as shown in Frasca et al. [41] due to which we only conduct experiments with SIGN for the purpose of comparison with LargeGT. In our experiments on `ogbn-products` and `snap-patents`, under the scalability constraint (-**δ**), *i.e.*, limited to 2-hop computations, SIGN-**δ** achieves 80.28±0.04 performance, comparable to LargeGT on `ogbn-products` but significantly underperforms with the score of 47.91±0.07 compared to LargeGT on `snap-patents`. Therefore, both SIGN and SGC, by extension, inherit the limitations of local-only models and exhibit poor performance in non-homophilic settings, such as `snap-patents`. LargeGT either matches or exceeds the performance over SIGN-**δ**.

## 5. Conclusion

In this work, we propose the LargeGT framework for designing Graph Transformers on very large graphs, such at those with at least millions nodes. We studied the characteristics and constraints necessary for a learning model to effectively scale on large graphs, namely model capacity, computational feasibility, and distributed training ability. We design LargeGT to satisfy the intended criteria, and show a proof of concept of how these components can be incorporated into a GT model to obtain competitive results in an improved runtime compared to architectures in prior literature.

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

# A. Additional Runtime Analysis

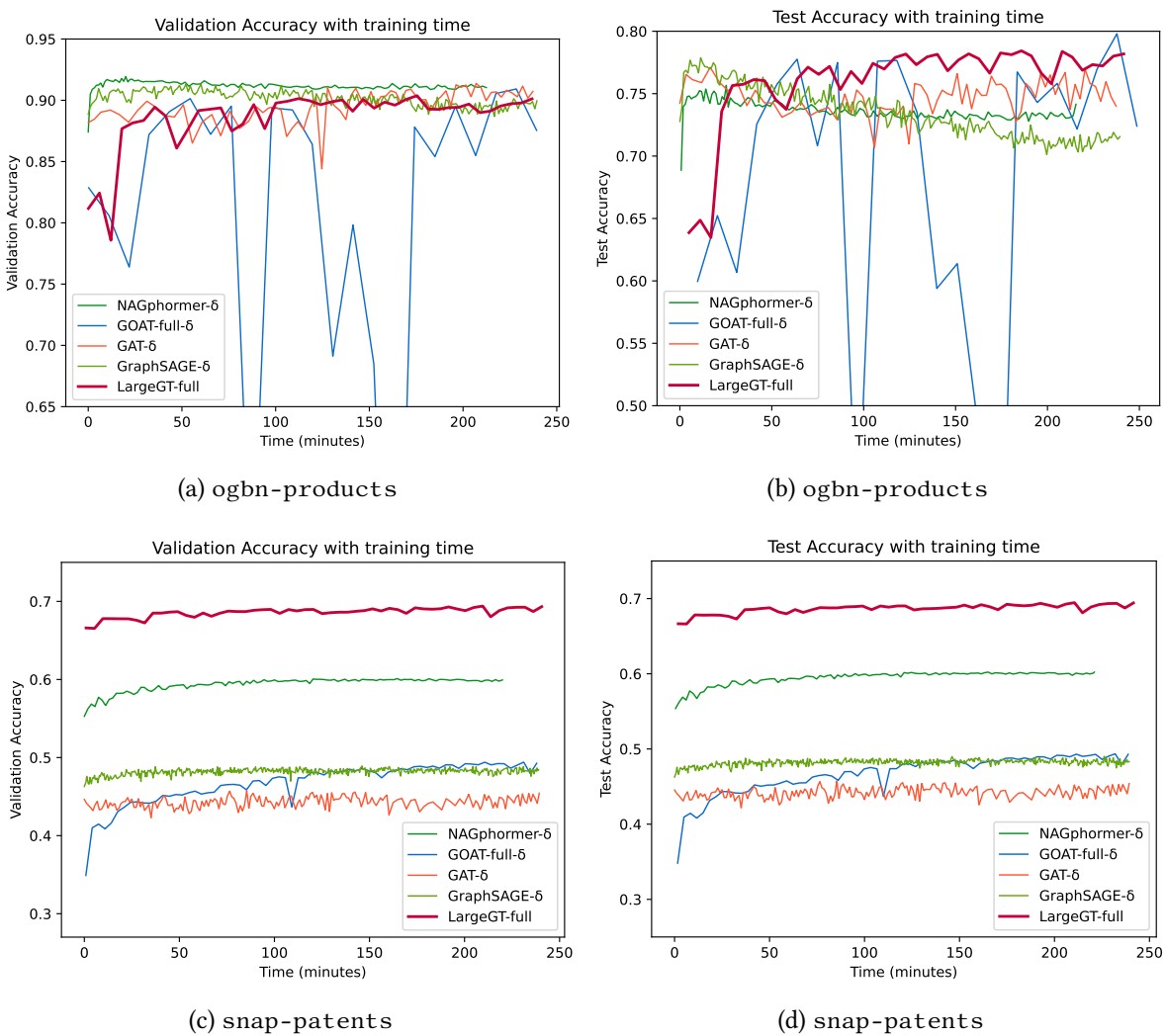

(a) `ogbn-products`

(b) `ogbn-products`

(c) `snap-patents`

(d) `snap-patents`

**Figure A.1:** Validation and test curves of the models with the running time in minutes.

Figure A.1 presents a convergence analysis of the scalable baselines with LargeGT with their validation and test performance for an unlimited number of epochs run within a computation budget of 4 hours (240 minutes). We observe better generalization curves of our proposed architecture against the baselines as the latter are largely affected by NS (neighbor sampling) while our architecture is based on the design principles which addresses some limitations of the NS used during training in the baselines. Additionally, recall from Section 4, we report a $3\times$ faster epoch time of LargeGT-full vs. GOAT-full-δ on `ogbn-products`. This is influential in the faster convergence of LargeGT as observed for `ogbn-products` in Figure A.1, where the generalization curves of LargeGT-full for $< 90$ minutes is similar to that of GOAT-full-δ for the entire 240 minutes, resulting in an earlier learning stability.

## B. LOCALMODULE

One of the main contributions of LargeGT (Section 3.2) in building localized representations is the introduction of a tokenization strategy that prepares a fixed set of input tokens for each graph node which is processed in the Transformer encoder of LOCALMODULE to produce local feature representations of the node. For this, first a set of $K$ local nodes are sampled offline using LOCALNODES Algorithm 1, which is then leveraged to prepared $3K$ tokens for each node during the mini batching step, following

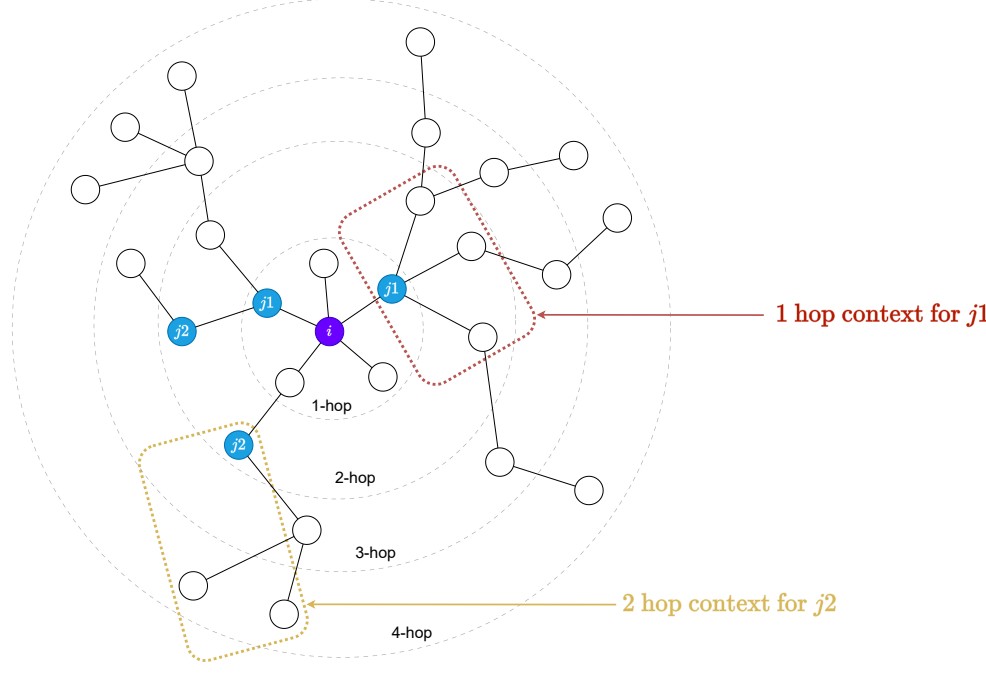

An Example Set of Input tokens for node $i$ for LocalModule at $K = 5$

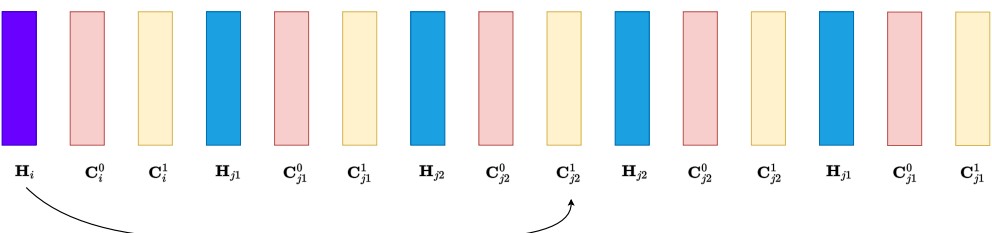

$\mathbf{C}_{j2}^1$ provides 4 hop information for node $i$ in consideration

**Figure B.2:** An example illustration of how the tokenization strategy using LocalNodes and InputTokens proposed in LargeGT which is input into LocalModule allows an effective receptive field of up to 4-hops for a node $i$ in consideration.

InputTokens Algorithm 2. The combination of the sampled nodes and the context features in Algorithm 2 effectively allows a receptive field of up to 4 hops for a node $i$ in consideration, using only 2 hop operations. This is illustrated in Figure B.2. We compute the 1 hop and 2 hop context features for all nodes using $\mathbf{C}^0 = \tilde{\mathbf{A}}\mathbf{H}$ and $\mathbf{C}^1 = \tilde{\mathbf{A}}^2\mathbf{H}$, respectively, where $\tilde{\mathbf{A}}$ is the normalized adjacency matrix and $\mathbf{H}$ is the node feature matrix.

In the figure, where $K$ is set to 5 for simple illustration, we can observe that there are $K - 1 = 4$ sampled nodes for the node $i$, where the nodes sampled from 1 hop are represented as $j1$ and from 2 hop are represented as $j2$. The 2-hop context feature for $j2$ will also form part of the token set for the node $i$, following InputTokens Algorithm 2. As such, the set of tokens of size $3K$ will include vectors which represent the a node $i$'s 4 hop neighborhood information.

## C. GlobalModule

The algorithm GlobalModule in LargeGT is adapted from [19] which uses a single layer Transformer encoder to allow every node in a mini-batch to global graph nodes, *conceptually* through a fixed sized $B$ centroids. The $B$ centroid tokens are stored in a codebook which is defined as $\mu \in \mathbb{R}^{B \times D}$ and are updated using an Exponential Moving Average (EMA) K-Means algorithm [19] at every mini-batch step. In Algorithm 3, $\text{MLP}_a$ and $\text{MLP}_b$ are two multi-layer perceptron modules with separate learnable

---

**Algorithm 3** GLOBALMODULE: Algorithm to output global representations of nodes in mini-batch

---

**Require:** Hidden features $\mathbf{H}_{\text{in}} \in \mathbb{R}^{M \times D}$ for $M$ nodes in a mini-batch, $\mu$ is the centroid computed by the K-Means algorithm, $\mathbf{P}$ is the centroid assignment index for each node.

**Ensure:** Return the output representations of all nodes in the mini-batch $\mathbf{H}^{\text{global}}$.

1: **for** $i = 0$ to $M - 1$ **do**
2:      $\mathbf{x} \leftarrow \text{MLP}_a(\mathbf{H}_i^{\text{in}})$
3:      $\mathbf{q} \leftarrow \mathbf{x}\mathbf{W}_Q$
4:      $\mathbf{K} \leftarrow \mu\mathbf{W}_K$
5:      $\mathbf{V} \leftarrow \mu\mathbf{W}_V$
6:      $\hat{\mathbf{H}}_i^{\text{global}} \leftarrow \text{Softmax}\left(\frac{\mathbf{q}\mathbf{K}^\top}{\sqrt{D}} + \log\left(\mathbf{1}_n\mathbf{P}\right)\right)\mathbf{V}$
7:      $\mathbf{H}_i^{\text{global}} \leftarrow \text{MLP}_b(\hat{\mathbf{H}}_i^{\text{global}})$
8:      Update $\mu$ using $\mathbf{x}$ through Exponential Moving Average (EMA) K-Means algorithm.
9: **end for**

---

parameters, while $\mathbf{W}_Q, \mathbf{W}_K, \mathbf{W}_V$ are the learnable parameters of the Transformer encoder.

## D. Extended Baseline Results

As mentioned in Section 4.1, we compare LargeGT with 'Scalable Baselines' which are constrained versions of existing MPNNs or GT baselines, denoted by Model-δ. The decision to compare LargeGT with 2-hop constrained versions of the baselines was to maintain a consistent scope of expensive computation incurred across all models. For instance, the cost of fetching $l$-hop neighbors for a node in a very large graph would be the same and agnostic of any selected model. Nevertheless, in this section, we provide an extended comparison of LargeGT with both 'Scalable Baselines' as well as their original models, for `ogbn-products`. Note that for the original models, we adopt hyperparameters in their respective papers. From Table D.1, we can observe that the best performing model remains the same while LargeGT, which only use up to 2-hop operations, remains better and competitive compared to the best performing model. Additionally, it is obvious that the 'Original Baselines' that employ greater than 2-hop operations are computationally demanding (upto $4\times$ in some cases), as compared to the 'Scalable Baselines'.

**Table D.1**
Extended results for `ogbn-products` where LargeGT is compared to Model-δ as well as the original baselines without the -δ constraint.

| Model | Original Baselines | | Scalable Baselines (-δ) | |
|---|---|---|---|---|
| | Test Acc | Epoch Time | Test Acc | Epoch Time |
| GraphSAGE | 78.17 | 22s | 76.62 | 7s |
| GAT | 79.21 | 86s | 77.38 | 31s |
| GT-sparse | 67.87 | 40s | 60.76 | 12s |
| NAGphormer | 77.71 | 22s | 75.28 | 8s |
| GOAT-local | 81.29 | 490s | 81.17 | 120s |
| GOAT-global | 70.28 | 3.5s | 70.28 | 3.5s |
| GOAT-full | 81.21 | 500s | 79.88 | 205s |
| LargeGT-local | | – | 78.95 | 45s |
| LargeGT-full | | – | 79.81 | 68s |

## E. Experimental Details

In our experiments, we use the baselines GraphSAGE [24], GAT [21], GT-sparse [11, 44], NAGphormer [18] and GOAT [19] to compare with our proposed architecture LargeGT. We follow the setup of the

**Table E.2**

Hyperparameters for `ogbn-products`

|  | GraphSAGE-δ | GAT-δ | GT-sparse-δ | NAGphormer-δ | GOAT-δ | LargeGT |
|---|---|---|---|---|---|---|
| batch size | 1024 | 512 | 1024 | 200 | 512 | 1024 |
| hidden dim | 256 | 128 | 128 | 512 | 256 | 256 |
| heads | – | 4 | 4 | 4 | 2 | 2 |
| pos enc | – | – | – | – | node2vec (64) | node2vec (64) |
| centroids ($B$) | – | – | – | – | 4096 | 4096 |
| NS | [20, 10] | [20, 10] | [20, 10] | – | [20, 10] | – |
| dropout | 0.5 | 0.5 | 0.3 | 0.3 | 0.5 | 0.5 |
| learning rate | 3e-3 | 1e-3 | 1e-3 | – | 1e-3 | 1e-3 |
| weight decay | 0.0 | 0.0 | 0.0 | 1e-5 | 0.0 | 0.0 |
| warmup steps | – | – | – | 1000 | – | – |
| peak learning rate | – | – | – | 1e-3 | – | – |
| end learning rate | – | – | – | 1e-4 | – | – |
| learning patience | – | – | – | 30 | – | – |
| hops | – | – | – | 2 | – | – |

**Table E.3**

Hyperparameters for `snap-patents`

|  | GraphSAGE-δ | GAT-δ | GT-sparse-δ | NAGphormer-δ | GOAT-δ | LargeGT |
|---|---|---|---|---|---|---|
| batch size | 1024 | 512 | 1024 | 200 | 2048 | 2048 |
| hidden dim | 256 | 128 | 128 | 512 | 128 | 128 |
| heads | – | 4 | 4 | 4 | 2 | 2 |
| pos enc | – | – | – | – | node2vec (64) | node2vec (64) |
| centroids ($B$) | – | – | – | – | 4096 | 4096 |
| NS | [20, 10] | [20, 10] | [20, 10] | – | [20, 10] | – |
| dropout | 0.5 | 0.5 | 0.3 | 0.2 | 0.5 | 0.5 |
| learning rate | 3e-3 | 1e-3 | 1e-3 | – | 1e-3 | 1e-3 |
| weight decay | 0.0 | 0.0 | 0.0 | 1e-5 | 0.0 | 0.0 |
| warmup steps | – | – | – | 1000 | – | – |
| peak learning rate | – | – | – | 1e-3 | – | – |
| end learning rate | – | – | – | 1e-4 | – | – |
| learning patience | – | – | – | 30 | – | – |
| hops | – | – | – | 2 | – | – |

**Table E.4**

Hyperparameters for `ogbn-papers100M`

|  | GOAT-δ | LargeGT |
|---|---|---|
| batch size | 2048 | 1024 |
| hidden dim | 768 | 512 |
| heads | 2 | 2 |
| pos enc | node2vec (128) | node2vec (128) |
| centroids ($B$) | 4096 | 4096 |
| NS | [20, 10] | – |
| dropout | 0.5 | 0.5 |
| learning rate | 1e-3 | 1e-3 |
| weight decay | 0.0 | 0.0 |

respective codebase of NAGphormer[4] and GOAT[5] to conduct the baseline experiments, while for the remaining, we follow their OGB implementation[6]. The hyperparameters of the experiments are as follows.

### E.1. Hyperparameters

[4]https://github.com/JHL-HUST/NAGphormer
[5]https://github.com/devnkong/GOAT
[6]https://github.com/snap-stanford/ogb
