# OpenReview forum: "Graph Transformers for Large Graphs"
_KDD.org/2024/Workshop/DL4KG — Submitted to DL4KG 2024_

### Official Review · Reviewer_RTSu · 2024-07-02
**Potentially interesting for graphs, limited impact for knowledge graphs**

**Rating:** 4
**Confidence:** 4

**Review:**

This paper introduces LargeGT, a novel Graph Transformer (GT) framework designed to address the computational challenges associated with large-scale node classification tasks. The authors propose a neighborhood sampling technique that is pre-computed, which is then combined with local and global attention mechanisms. By conflating 1- and 2-hop neighbourhoods in their sampling approach, the method is able to go up to 4-hops with less computational demands. The local and global attention mechanisms are adapted from state-of-the-art works.
LargeGT is evaluated on three popular node classification benchmarks, and compared to a few relevant baselines, which the authors slightly adapted to consider fewer neighbours and thus be more scalable (although this was not necessary in many cases and results in decreased performance for the baselines). LargeGT surpasses baseline performance in 2 out of 3 datasets and is faster than the best-performing baselines.


My major concern with the paper is that it is not a very good fit for the workshop. It focuses only on GNNs and graphs, and does not specifically address KGs.
While scalability is definitely a challenge for GNNs over KGs, the work does not particularly focus on KGs nor on the more standard KG completion task (link prediction). Interestingly, OGB does provide a very large Wikidata link prediction challenge: WikiKG90Mv2, a Knowledge Graph (KG) extracted from the entire Wikidata knowledge base.
This dataset is part of the OGB Large Scale Challenge datasets. https://ogb.stanford.edu/docs/lsc/, which are definitely a more suitable challenge to the proposal of the paper.
While this is a major flaw --- and behind my less positive rating --- I leave below some suggestions and discussion points.

The proposed approach is methodologically sound, with its major contribution being a smart approach to offloading the issues of graph scalability to a pre-computation step that determines the context of each node based on a sampling of the neighbourhood. The other aspects of the methodology are based on existing approaches.
However, the implications of the neighbourhood sampling are not properly discussed or analysed. LargeGT conflates the 1-hop and 2-hop neighbourhoods. It would go a long way towards understanding the impact of this step to perform studies that compare defining a neighbourhood at just 1-hop with 1- and 2-hop.


The experimental design is somewhat limited.

First, as stated above, the benchmark datasets selected are not the most suitable. Moreover, it is unclear why LargeGT is focused only on node classification.

Second, the choice of baselines does not include the best performing systems in the OGB leaderboards for the selected benchmarks. While authors state “The goal of our experimental setup is to show how the scalability constraints affect existing models’ capabilities, which can be addressed by LargeGT; for this reason, we do not use enhanced techniques for obtaining top leaderboard results such as auxiliary label propagation [44, 67] or augmentations [68].”, the fact remains that LargeGT would rank 44th on OGB leaderboard for ogbn-products and 15th for ogbn-papers100M. The less than stellar results on these benchmarks should be discussed in more detail.

Third, the choice of “constraining” the baselines to ensure they are scalable is questionable. The text is very unclear on what these constrained versions are, which is only briefly explained in a table caption. But more importantly, some of the models used as baselines are able to run fully on the datasets. The GOAT paper includes results for both ogbn-products and snap-patents, and in the first case the results surpass those of the constrained version (82.00 ± 0.43). The ogbn-products leaderborad has GraphSage at 0.7850 and GAT with neighbour sampling at 0.7945 (which would be 3rd place behind GOAT). I do not think it is a fair evaluation to "constrain" models that do not appear to need it. The standard results are presented in the appendix, but by showing only constrained versions, LargeGT appears to perform better than it actually does. Moreover, LargeGT can access a neighbourhood of up to 4-hops, while others were constrained to only 2.

Finally, the analysis on computational time is not entirely fair. Since LargeGT pre-computes the neighbourhood sampling, the time spent on this task is not considered for model training. However, this aspect should definitely be discussed. The reader must be made aware if the neighbourhood sampling is very costly to run, and since you cannot train the model without it.

The paper’s title is a bit too broad in meaning: Graph Transformers for Large Graphs. There are several methods that could be described by this title.

---

### Official Review · Reviewer_AzC2 · 2024-07-02
**Merging local and global features while maintaining scalability in the context of graph transformers**

**Rating:** 4
**Confidence:** 4

**Review:**

Overview: The paper highlights the limitations of existing graph learning models in integrating local and global graph features, especially when dealing with very large graphs. It proposes a method that effectively merges both local and global features while maintaining scalability. The proposed method involves pre-computing neighborhood sampling and utilizes a model that combines a graph transformer (for local features) and a global codebook (for global features). The authors claim that their model demonstrates good performance and significantly reduces training time compared to existing methods.

Relevance: Learning on large graphs is a highly relevant topic as many models struggle to scale effectively. The paper addresses important motivations. → (5/5)

Position with respect to SOTA: The approach is not compared to state-of-the-art (SOTA) models but only against simplified versions of baseline models. The authors explicitly exclude SOTA models, citing their primary goal of evaluating scalability. We do not consider this a valid justification. → (1/5)

Novelty: The integration of local and global features in the proposed model might be novel. The second contribution appears to be the pre-processing of neighborhood sampling, which seems more like an engineering improvement rather than a novel research contribution. Further, the neighborhood sampling is not sufficiently described. → (2/5)

Potential Impact: While the topic is highly relevant, the paper does not provide a substantial research contribution. → (2/5)

Text:
The paper is difficult to follow and contains numerous typos. The first three pages focus excessively on motivation, with design principles repeated in the introduction and the "Recipe for Building Transformers for Large Graphs" section. This redundancy could be reduced to allow more space for explaining the foundational concepts and model architecture. The functionality of the global codebook, a critical component of the model, is not clearly explained. Additionally, the algorithms are inadequately described.

The paper repeatedly claims a 16.8% performance gain on ogbn-products and snap-patents, which contradicts the values reported in Table 2. Unfortunately, only accuracy (Acc) is reported, with no mention of F1, Precision, or Recall.

The claim that the proposed model is three times faster than existing models is not substantiated by the figures. Figure 2, which plots epoch time against test accuracy, is not a representative measure of overall training time and speedup, as different models require varying numbers of epochs. Moreover, Figure A.1 (a) and (b) suggest that the proposed model trains slower than existing ones, while only (c) and (d) show a significant improvement. Moreover, the claim of faster training cannot be concluded from these plots.

Open Questions:

In the abstract: “finding the optimal trade-off between speed and accuracy with sampling techniques remains challenging” → What does accuracy mean in the context of sampling?
It is unclear what features are provided to the local transformer model. Are node IDs or other node features included?
In the Introduction, under "Local Representations": “We represent a novel tokenization strategy that prepares a fixed set of tokens for each graph node to be processed by a Transformer encoder.” This is never explained in the text. What does this mean?
It is unclear how performing neighborhood sampling in a preprocessing step reduces memory requirements. The sum of sampled neighborhood graphs should be larger than the total graph. Even if these neighborhoods can be distributed across multiple nodes, the benefit is questionable. Moreover, this design decision lacks practical evaluation on a computing cluster with multiple nodes.
“For simplicity, we will represent H_i in the following equations” → What does this mean? Furthermore, H_i^in is not consistently replaced by H_i.

---

### Official Review · Reviewer_GNZZ · 2024-07-03
**Carefully written, interesting contribution**

**Rating:** 8
**Confidence:** 3

**Review:**

The paper presents an approach to train transformer models on graphs in combination with sampling.

The paper is well written, with extensive related work and carefully designed notation. The method is competitive, but not state-of-the-art. Overall, I think the contribution is interesting, and the paper should be accepted.

However, I have the following questions and comments that the chair may not, and that the authors should strive to address.

1) The paper does not specifically address knowledge or knowledge graphs in any way (only general graphs). This puts it outside the call for this workshop as it is on the website. I leave it to the chair to decide whether the paper is in scope, or whether a sufficiently interesting out-of-scope paper, may still be a nice addition to the workshop.
2) Despite the extensive literature list, some methods should still be mentioned and compared against (at least conceptually if not experimentally). For example, LADIES does not appear to be cited, and FastGCN is only mentioned in passing. The proposed method is very similar to these more general sampling methods, and it would be good to see explicitly what the differences and similarities are. For instance, if I took two layers of LADIES sampling (which would sample randomly from a 2-hop neighborhood), and then added a transformer layer on top as the MPNN, how close would I be to the LargeGT network? I guess the key difference is the global update, in combination with sampling, but that would be good to spell out.
3) It isn't clear to me why the section "On comparison fwith SGC and SIGN is kept separate from the rest of the results. It seems that these numbers could easily be added to table 1. Placing them in the running text of a separate section gives the appearance of trying to hide them (which I'm sure isn't the idea, since they aren't damaging to the general message of the paper).

If issues 2 and 3 are properly addressed, and issue 1 is fine with the chair(s), I think the paper would make a fine contribution.

For minor points: the language is readable, but the authors often skip articles like "the" and "a" showing that the paper was not written by a native speaker. It would be to address these issue. As a personal pet peeve, I strongly suggest replacing all occurrences of the verbs "utilize" and "leverage" with "use".

---

### Decision · Program_Chairs · 2024-07-09

Reject